# Mothering a Child with ADHD in the Ultra-Orthodox Community

**DOI:** 10.3390/ijerph192114483

**Published:** 2022-11-04

**Authors:** Jennifer Budman, Adina Maeir

**Affiliations:** School of Occupational Therapy, Hebrew University of Jerusalem, Jerusalem 9190501, Israel

**Keywords:** vulnerable populations, health-related quality of life, ADHD

## Abstract

Mothers of children with ADHD are at risk for negative health outcomes. The socio-cultural environment and everyday experiences in life roles may influence psychological health and quality of life. The ultra-orthodox Jewish (UOJ) community is an insular community who is underrepresented in the research, and as such there are no studies exploring the experience of mothering a child in this community. Thus, this study aims to explore the lived experience of mothering a child with ADHD in the UOJ community. The study used a descriptive qualitative phenomenological approach. Ten UOJ mothers of children with ADHD underwent semi-structured in-depth interviews to explore their lived experiences of mothering a child with ADHD. Thematic analysis was carried out on the transcripts. Four main themes, along with several subthemes, emerged from the qualitative analysis: child ADHD manifestations, maternal role, social factors, and self-care. Unique characteristics of the UOJ culture were apparent throughout the themes. UOJ children with ADHD display similar manifestations of symptoms in daily life to those in the general population and maternal burden is similarly present. However, unique perceptions of their maternal role, social factors, and legitimacy for self-care shed light into the impact of this culture on their lived experience. Findings may help promote culturally sensitive health care and interventions for this understudied population.

## 1. Introduction

Attention deficit hyperactivity disorder (ADHD) is a chronic neurodevelopmental health condition characterized by a persistent pattern of inattention, hyperactivity, and impulsivity [1]. The prevalence of ADHD in children is estimated at 5–7% worldwide, making it one of the most common psychiatric disorders in this population [2]. The behavior manifestations associated with ADHD place tension on the family unit, negatively affecting the function among members [3]. Mothers are more often the main caregivers for children, and as such, a significant portion of the care involved in raising a child with ADHD falls on them [4]. Mothers of children with ADHD (MoCwADHD) experience physical [5], psychological [6], social [7], and functional [2] burdens, which has been shown to negatively impact their psychological health [6] and quality of life (QoL) [8].

Factors that have been associated with the severity of negative psychological health and QoL outcomes in MoCwADHD include child factors such as the severity of the child’s ADHD symptoms [9] and maternal factors such as parental self-efficacy [10]. Environmental factors have also been associated with negative psychological health and QoL among MoCwADHD. For example, one’s immediate social and cultural environment can influence beliefs regarding illness and treatment, can influence health decisions and impact accessibility to treatment [11], and can influence understanding the mothering role [12,13]. Thus, the psychological health and QoL of MoCwADHD is not only influenced by having a child with ADHD and personal factors but also by the social and cultural environments in which they were raised and currently live [14].

Culture is referred to as the “shared ideas, beliefs, systems of concepts and meanings, values, knowledge, ways of being, customs, and often, language that arise over time within a particular group” [15]. Understating the role of culture is important to promote inclusiveness and client-centered health care among diverse populations [16]. Cultural competency is a contextual and dynamic process where health professionals adjust their practice to meet the unique cultural sensitivities and needs of clients with understanding and communication, and includes cultural awareness, knowledge, and skills. Studies have emphasized the need for cultural competency regarding the ultra-orthodox Jewish (UOJ) population to support building effective therapeutic relationships and interventions for this community [17].

UOJ are a group of highly religious individuals that belong to an internally cohesive and segregated community where values, beliefs, and behaviors are greatly influenced by their society’s cultural codes. They are an insulated community with a commitment to orthodox Jewish laws and customs based on the Torah, the first testament, and rabbinical authority, which influences all aspects of daily living. UOJ tend to have larger than average families and lower than average incomes. They display cultural conservatism, along with fixed boundaries between themselves and the general population to minimize outside influences, and thus have less accessibility to the internet and treatment resources [17]. In this community, a woman’s highly valued role is being a mother. Motherhood is viewed as a religious responsibility connected to a woman’s relationship with God and a means for self-actualization. Children’s behavior is viewed as the mother’s responsibility, and challenges relating to children’s disobedience may be considered a reflection of her capabilities in raising the next religious generation [13]. Furthermore, this community highly values religious education and learning, which is in disaccord with the cognitive challenges displayed by children with ADHD [17]. These values are also manifested in the social criteria set forth by the community to rate individuals at various life junctions, such as finding a marriage partner and getting accepted to the educational institutions [18]. While the prevalence of ADHD is generally reported to be lower in highly religious communities [17,19], there is research that indicates that the rate of children with ADHD in the UOJ community is similar to that of other countries [20]. UOJs are the fastest growing population in Israel and one of the fastest growing populations in the world. They account for approximately 10% of the overall population in Israel and are estimated to triple in growth and comprise 30% of the total population in Israel over the next three decades [21]. As members of this community generally keep to themselves and under-utilize public mental health services, for fear of stigma and lack of culturally sensitive treatments [22], there is little knowledge in the literature regarding the experience of mothering a child with ADHD in the UOJ community.

### Aims and Objectives

Considering the known health risks of mothering a child with ADHD, the UOJ community was chosen for this study due to several unique factors. The religious responsibility attributed to the mothering role, the exceptionally high social value placed on religious education, as well as the disparity of health services, may pose additional challenges to mothering a child with ADHD in this community. Therefore, the present study aimed to explore the lived experience of mothering a child with ADHD in the UOJ community to support cultural competency among health professionals.

## 2. Materials and Methods

### 2.1. Design

This study used a qualitative descriptive design with a phenomenological approach. The qualitative descriptive design is a comprehensive depiction of specific events experienced by individuals or groups of individuals [23]. The phenomenological approach describes the essential structure of a phenomenon, which in this study is the lived experience of mothering a child with ADHD in the UOJ community and can only be understood through the descriptions of the participants [24].

### 2.2. Participants

Participants included ten UOJ mothers of children between the ages 6–18 years old, diagnosed with ADHD by a licensed medical professional. All the mothers were married, aged 30–45 years old. Most mothers (*n* = 7) completed a post-secondary education or higher and four mothers reported income below the national average. The number of children per family ranged from 3–6, with half the mothers reporting to have 6 children. Three mothers reported having more than one child with ADHD and most of the mothers (*n* = 7) stated their child with ADHD was male. Most of the mothers (*n* = 6) displayed mild to severe psychological symptoms. Parenting stress ranged from 28–35, moderate to high. Four areas of QoL were reported: physical (range 39–75), psychological (range 35–80), social (range 37–80), and environment (range 54–76). One mother reported having ADHD and child ADHD index ranged from 45–88. For full details of mothers’ background information see Table 1 and Table 2.

### 2.3. Data Collection

Study was approved by the institutional review board (#20122020). The study was performed in Israel and was advertised in Israeli outpatient clinics through the country and social media groups, after receiving permission from clinic and group administrators, and the snowball method. The advertisement contained an explanation of the study, both authors’ credentials and the first author’s contact details. The first author is a PhD student and an occupational therapist who is a member of the UOJ community, and the second author is a researcher in ADHD and QoL. An initial phone call with each respondent ensured participant’s suitability for the study and provided assurance of anonymity. Written consent was obtained from all mothers prior to start of the study. The interviews were conduction from December 2020 through to March 2022.

Prior to the interview, participants in the study completed a background questionnaire including socio-demographics, child ADHD symptomatology, and maternal psychological health and QoL information. Mothers were provided the option of receiving questionnaires by email or standard mail and all chose to receive by them by email. Following the return of questionnaires, the first author arranged to meet each mother individually at a place of their choice to conduct an interview, as the appropriate method to explore and identify the mothers’ lived experiences [24]. The interview guide was developed by both authors and based on an extensive literature review regarding the lived experience of MoCwADHD in the general population. The interview guide can be found in Table 3. One interview took place in the mother’s home, three in the author’s clinic and six over zoom. The interviews lasted approximately 60–90 min. Sample size was based on saturation according to the definition of conceptual depth, where the data collection stops when the researcher has reached sufficient depth of understanding and was within the acceptable range of 6–12 interviews [25,26]. Design and reporting of this study were according to the consolidated criteria for reporting qualitative research (COREQ) checklist [27].

### 2.4. Measures

#### 2.4.1. Semi-Structured Qualitative Interview

Interview questions were based on the extensive literature regarding the lived experiences of mothering a child with ADHD in the general population [4,7,8], and the relationship between culture and the mothering role [13]. Thus, open-ended questions were developed to try and capture the MoCwADHDs’ experiences within the context of her socio-cultural environment.

#### 2.4.2. Background Questionnaire

Demographics: including mothers’ age, family status, number of children, number of children with ADHD, age of child(ren) diagnosed with ADHD, household income status, level of education completed, and self-report of whether mother is diagnosed with ADHD.

Child ADHD symptomatology: Conners Third edition (Conners 3) [28], ADHD index was used to quantify the severity of the child ADHD. Items were measured on a 4-point scale (0 = not true at all to 3 = very much true). Raw scores were converted to T-scores. Higher scores represent more severe symptomatology. The measure has good internal consistency coefficients, high test-retest reliability, and effective discriminatory power [28].

Parental stress: Parenting Stress Items (PSI) [29] measures parental stress by asking parents to consider how “tense” or “frustrated” they feel about parenting their children. Includes 11 items scored on a four-point scale (1 = not at all to 4 = very much so), yielding scores ranging from 11–40, with higher scores indicating greater levels of parenting stress. The PSI is a validated and reliable measure that has been used to measure parenting stress among parents of children with ADHD [30].

Psychological health: Patient Health Questionnaire-4 (PHQ-4) [31]. Four-item self-report questionnaire consisting of a two-item depression scale and two-item anxiety scale. Items are measured on a 4-point scale (0 = not at all to 3 = nearly every day), yielding scores ranging from 0–12, with higher scores indicating greater psychological distress (0–2 = none, 3–5 = mild, 6–8 = moderate, 9–12 = severe). The PHQ-4 displayed good internal consistency and validity as a measure of depression and anxiety symptomatology [31].

Quality of life: WHOQoL-BREF [32] assesses self-perception of QoL in the context of the culture and values systems in which one lives and in relation to one’s goals, expectations, standards, and concerns. This includes 26 items that assess four domains, including physical health, psychological health, social relationships, and environment. Scores range from 0–100, with higher scores representing higher levels of QoL. The WHOQOL-BREF has displayed good discriminant validity, internal consistency, and test–re-test reliability. It has been used in research on MoCwADHD [33].

### 2.5. Data Analysis

Interviews were recorded and transcribed verbatim by the first author, who then read and re-read the transcriptions to become familiarized with the data. The interviews were conducted in Hebrew. The excerpts included in this article were translated into English by both authors who are proficient in both English and Hebrew. ATLAS.it 9 software (ATLAS.ti Scientific Software Development GmbH, Berlin, Germany) was used to organize, analyze, and visualize the data. Throughout the entire process, the authors were sensitive to any bias that may occur considering that the first author is a member of the UOJ community. This was addressed by analyzing the text using the principles of thematic analysis [34], where there are constant discussions regarding the data and comparisons between the authors’ analyses, at each stage of the analysis. Both authors first identified initial coding of meaningful units of the interviews, and then compared codes. Discrepancies lead to a discussion before final codes were established. The first author then collated these codes into potential themes and sub-themes. Through constant reference to the data, the themes and sub-themes were refined several times, ensuring that they made sense in relation to the codes and the entire dataset. The authors conducted discussions regarding the themes and sub-themes until agreement was established. This led to four key themes with their subsequent sub-themes. A coding tree was generated to visualize the main and sub-themes. The final stage of analysis involved the selection of relevant quotations from the dataset, which represent the lived experience of mothering a child with ADHD in the UOJ community.

## 3. Results

Four themes emerged from the analysis of the mothers’ data: Child ADHD manifestations, maternal role, social factors, and self-care. The themes are further broken down into sub-themes, presented in Figure 1, and are explained and substantiated with representative examples of verbatim quotations from the interviews.

### 3.1. Child ADHD Manifestations

When mothers were asked to describe their child, they stated examples describing cognitive, behavioral, and emotional manifestations of their child’s ADHD. All the mothers described that their child has difficulty staying focused and concentrating when talking to them, as well as in daily tasks:

‘He doesn’t succeed in sitting down, in concentrating… when I talk to him, after one sentence he loses me entirely, and all sorts of things like that’.(YN)

‘She does a million things, jumps from thing to thing’.(G)

‘While eating, he’s very scattered, very distracted, going from thing to thing, suddenly playing, he doesn’t persistent in any activity’.(BC)

The child’s challenge in regulating their emotions and behaviors according to social norms was also described by all the mothers:

‘He can cry, he really can… sometimes it’s like he gets into a state of anger, and when he’s angry then he can throw things’.(H)

‘He fights a lot with his siblings, last week there were murder-style hitting episodes, he beats them up, my G-d’.(Y)

‘He doesn’t have boundaries, honestly, he has no boundaries. He doesn’t know when to stop, what’s appropriate and what isn’t’.(H)

‘The way she sits, there are a ton of people in the living room, she wants to lie down now, so she will lie down on the couch. It does not even matter that there are a ton of people in the living room and so it’s inappropriate, and it is expected that she will behave differently’.(M)

### 3.2. Maternal Role

The maternal role was subdivided into the following themes: Perception of mothering, burden of mothering a child with ADHD, and resources and strategies.

#### 3.2.1. Perception of Mothering

Mothers expressed both their perceptions regarding the mothering role in general and specifically to a child with ADHD. The maternal role was expressed by all mothers as the most central role in their lives, and they repeatedly stated themselves as being a “mother first”.

‘The most important thing for me is to be a mom. I try as much as I can to give up everything possible so that my role as a mother is fulfilled properly’.(Z)

The centrality of the mothering role was expressed by how responsible they feel for the totality of their children’s needs:

‘To be attentive to the children, to see their needs, understand them, both their emotional needs as well as their physical needs, everything, like to be theirs, their everything, this is most important to me’.(T)

‘To learn to play all the roles (for the child) because each child is different, lots of patience, knowledge, responsibility’.(G)

Religious framing was also expressed by all the mothers when describing their mothering role:

‘I think that Hashem (God) gave us children and every child is his own world. As the sages say—‘Educate the child according to his way’. Each child has his own way’.(Y)

‘I really enjoy it, I really love home, I love the family… I look forward to Friday and Shabbat (Holy seventh day), the atmosphere of Shabbat, the kids at home, it’s all for me really, so I really like being a mom’.(T)

#### 3.2.2. Burden of Mothering a Child with ADHD

Mothers expressed the challenge of mothering a child with ADHD as all-encompassing, affecting themselves and their families, and as an intense lifelong struggle:

‘It’s something that is constant, it’s challenging, constantly thinking, thinking how to help him, how also not to let him manage us because after all we also have the life of each one, mine, my husband’s, his siblings’.(H)

‘I need to be in control, not let him try to manage me… naturally you get angry and upset and lose patience, so it takes a lot’.(BC)

‘I think the burden is in managing the family with children with ADHD because it can get the whole family off track’.(G)

I can very much see how it can destroy a marriage’.(G)

‘With ADHD you can’t say, ok that’s it, it’s not a struggle that you start with, and you know where it ends’.(Z)

Half of the mothers also described their feelings of incompetency in their mothering roles:

‘I need to prepare her for life. And if I don’t prepare her properly then I failed in my role’.(M)

‘I don’t know, I hope that I am good but for sure I make a lot of mistakes’.(G)

All the mothers described having been engaged in a difficult process of accepting their child with ADHD. The mothers expressed the disparity between harboring normative expectations alongside their child’s real difficulties:

‘We had to open our heads a lot to accept them. On the one hand they are normal children, on the other hand, they have a lot of difficulties. So, it is very hard because it is not a child in special education, that you know there is definitely something wrong with them, and then you accept it more. On the one hand there are a lot of expectations for the child to be normal but there are their difficulties, so I think that is the hardest actually’.(G)

‘I am a person that when I start something, I finish it. So, seeing this (child with ADHD behavior), it’s very different from my character. This is the main thing that is very difficult for me’.(Y)

And some mothers went so far as to express feelings of rejection towards their child:

‘I said “No, this is not my child, and I know he has so many good things and it’s hard for me, it’s hard for me to remember that’.(H)

Most mothers expressed being able to eventually accept their child:

‘First of all, it means accepting him as he is, this is the first thing, because yes there are things I would like maybe otherwise. Learn to see the virtues. I see my children with ADHD as brilliant, that they are brilliant, social, and very successful, so first it is to learn to see’.(T)

The difficulty of mothering a child with ADHD was also related to the perceived ambiguity of the disorder:

‘We didn’t really know what it is. Because his disorder is not a disorder… it is not a disorder with hyperactivity as we know with ADHD, it is ADD, it is just ADD… So, we didn’t know to put our finger on the point’.(YN)

‘But I also saw this in other boys that… it’s not so nice, but that’s boys’.(H)

‘Nutrition, I know that sugar can makes them bouncier. Apparently sugar and flour affects it’.(H)

Considering this ambiguity, receiving a diagnosis was expressed as helpful in attributing their child’s behavior to the diagnosis and in understanding the reason for their child’s behavior:

‘I often felt he was just not listening to me. And really until we got the diagnosis, I then saw that he was not really able to listen to me—he could not always listen to me. He is distracted, he is in his own world sometimes’.(YN)

#### 3.2.3. Resources and Strategies

Mothers described various coping strategies when dealing with their child’s ADHD and the need for more resources. Sub-themes included pharmacological treatment, management skills, psychological re-framing, avoidance, and the need for more resources.

Pharmacological treatment. Pharmacological treatment was expressed as a positive resource by most of the mothers:

‘When I gave him the Ritalin, we suddenly discovered another child. Suddenly he was attentive, and suddenly he sat and was quiet. It calmed us down’.(H)

A few mothers expressed ambivalence in giving their child medication. They expressed being unsure of their own biased intentions and if it was really the correct thing to do for their child:

‘I feel like I don’t know if I want to do it (give medication). I’m constantly debating… maybe it’s for me, maybe it’s a little selfish, but I’m also thinking about him’.(YN)

And one mother described the negative effect that medication had on her child:

‘He is not really happy… I really feel for him that he no longer has the joy of life he once had’.(Y)

Management skills. Mothers described skills that helped them, such as having a routine, supportive communication, on-line assessment of the situation, and taking it one day at a time:

‘I also found some kind of technique for the mornings. I wake the girls first, I come to him—“M, you have another 10 min to sleep”. Let him stretch a little, move his muscles, whatever he wants’.(R)

‘Pleasant ways, good conversations, empowerment, I feel that punishments and anger are things that work less with him’.(Zh)

‘With him I’m learning that there is no black and white, you have to be attentive and see according to his mood, constantly accept and be gentle with the situation and do mapping, like what is better, what isn’t so good, which approach is better… I definitely feel that the skills I have developed for myself is really something that is very helpful to me’.(Zh)

‘First of all, it’s the ability to spot things a few minutes in advance so you can avoid them’.(Zh)

‘With these kids one has to really think about the next day and not the coming year. And not to think about what will happen far in the future… How will he manage in class, how will he manage at his bar mitzvah? I’m trying to think less about it (the future)’.(YN)

Psychological re-framing. Most of the mothers used reframing, within their religious context, viewing the experience of mothering their ADHD child as a growth opportunity:

‘I feel as a mother that this is another challenge I need… that Hashem (God) has given me, and we will get through it’.(YN)

‘I chose to take this crisis and grow from it rather than fall from it. I am learning from it, and it is building me’.(Z)

‘I think it could have also been that because of M (daughter), I am who I am, what I am, what I want. And I think this made a big difference (for me)’.(M)

Avoidance. Some mothers described the challenge as too great, that they just wanted to avoid it:

‘Sometimes I feel it is too much for me and I want to run away from it’.(BC)

‘I haven’t returned back to the doctor, I didn’t listen to what he said and I can’t deal with it’.(H)

Need for more resources. Despite the various resources and strategies described above, all the mothers expressed their experience of having insufficient resources to cope with their child’s ADHD manifestations at some point during their interviews and their need for more tools:

‘What could have helped me is how to help him, that is mostly what I think. How to help him become more independent, how to manage on his own’.(BC)

### 3.3. Social Factors

Mothers described the various social factors that influence their role as mothers and, more specifically, mothers of children with ADHD. The social factors that were described were divided into the following sub-themes: cultural role of breadwinner, ADHD stigma, support from partners and family, the healthcare system, and the educational system.

#### 3.3.1. Cultural Role of Breadwinner

It is common for women to be the main breadwinner in the UOJ community to allow men to study religious texts full-time [35], and as such, most of the women were the main breadwinners in their families. The experiences of their social role as breadwinners were expressed with both negative and positive aspects:

‘I work hard… I am the breadwinner’.(T)

‘I work very hard, I’ve always worked, since I got married, it takes a lot’.(G)

‘I can really say that I work very very hard and pay a price for trying to combine these two things (motherhood and working)’.(Zh)

‘I receive self-fulfillment and satisfaction (from work); I have good abilities to empower the team that works with me. I provide them something. I have lots of devotion and patience. Bottom line, I need the income, it’s (the income) important, but I could not have done this (work) without everything (the self-fulfillment and satisfaction) I mentioned before’.(Z)

‘I worked with flowers, and it did good for me. It gave me fulfillment, creativity, it helped bring out my creativity. I felt good with it, that I am good, that I am good at it’.(H)

#### 3.3.2. ADHD Stigma

Culturally specific ADHD stigma was expressed by mothers as an influencing factor on their mothering experiences of children with ADHD. Most of the mothers brought up the ‘shidduch’ at some point in the interviews. The ‘shidduch’ (marriage match) is the process of finding a suitable marriage partner in the UOJ community. Boys and girls are kept separate from an early age to ensure modest behavior between the sexes. When children become of marriageable age (18–20 years old for girls, and 20–21 years old for boys) then a match maker intervenes to set them up according to predetermined religious and social criteria set forth by the parents and the community in which the family lives. The ‘shidduch’ is often seen as a conformation of social status in the community [36]. Some mothers expressed how they kept their child’s diagnosis a secret as to not negatively influence the ‘shidduch’ in the future:

‘I’d rather not say anything, I want to protect his honor, mainly for the shidduch (marriage match)’.(Y)

While others explicitly expressed how they think their child with ADHD has the right to a good ‘shidduch’ despite their diagnosis:

‘Even a child with ADHD deserves to marry an excellent girl and he can achieve this’.(R)

One mother also stated that the stigma of having a child with ADHD is changing in her community:

‘This (ADHD) is almost normative. So, I don’t feel different or strange’.(YN)

#### 3.3.3. Support from Partners and Family

Mothers expressed various degrees of support they receive from their spouses and other close family members. Most of the mothers used words such as “we” when describing parenting their child, referring to their husbands’ active involvement and participation in parenting their ADHD child, however others reported having less support:

‘I have help from my husband, and we honestly try to do everything so it will be good for him (our son), so it will be good for us’.(Zh)

‘I have little support from my husband, from my mother-in-law, little support, emotionally and physically’.(BC)

#### 3.3.4. The Healthcare System

Mothers also described their experiences in the healthcare system regarding the management of their child with ADHD, in the context of their culture. Almost all the mothers described not receiving professional attention and service. However, one mother described her experience as supportive and attentive to her specific needs: 

‘We went to Dr. … and straight away she gave a diagnosis, based on a short questionnaire filled out by the Rebbe (teacher). Straight away she said, in this situation…I’m giving it (medication) to him. I felt like she was saying, whatever, you want I will give you; you want 100, I’ll give you 100, you want 20, take 20. That is how I felt (dismissed) in the process with her’.(R)

‘More or less the experience was very good, I have had only one bad experience in recent years, very pleasant doctors, very informative. The health care system for the parent has been a very pleasant experience, the doctor thinks about the home environment and if you need help at home’.(Z)

It was also expressed that treatment suggestions by the healthcare system were often culturally inappropriate:

‘He (the doctor) highly recommended psychological treatment for him. And we haven’t started yet. I don’t know these psychologists. I wanted to check out a psychologist who is appropriate for our community’.(H)

#### 3.3.5. The Educational System

When describing the educational system, mothers expressed an overall awareness of ADHD. However, medication was encouraged as the main treatment option, especially in the boy’s schools. There were gender differences found between the support the mothers felt their child with ADHD received from school, with positive experiences expressed for the girls over the boys. Mothers of boys expressed that the schools did not have sufficient psycho-educational tools to manage their children’s ADHD symptomatology, especially considering the highly demanding learning environment which includes long hours of studying religious texts that require high concentration and focus [37]:

‘Their (teachers) reaction is to give Ritalin so that the child will sit quietly, so he will have more concentration’.(Z)

‘I do not feel that there is a Talmud Torah (boy’s primary school) that can really give him everything he needs. The rabbis who teach in the Talmud Torah do not have professional training to teach even ordinary children, they have not really learned, they don’t have didactic tools, practical tools how to educate… even more so, for a child who needs a little more’.(Zh)

One mother who has both a son and daughter diagnosed with ADHD stated:

‘I think it depends where my children learned. Yes, the school was supportive (for the daughter), but by the boys, hmmm, it was actually better at my daughter’s school’.(G)

Two mothers of girls with ADHD, described similar experiences regarding the support their daughters received from school:

‘They gave her the best she could get, she studied at Beit Yaakov. It’s an amazing team and lots of help also beyond the studies themselves. Lots of emotional support, lots of knowledge, and lots of empowerment… The staff there demands more from her. It raises her, empowers her, gives her strength, ability’.(M)

Some mothers, regarding their sons with ADHD, expressed choosing to give up on their religious educational ideology so that their child can receive adequate support for their needs:

‘We are looking for a first grade and because he is a boy, he needs a cheider (UOJ boys’ primary school), and there is not really a cheider who will give him the tools he needs. So, we will probably have to send him to a place that’s a little less religious than we are’.(YN)

### 3.4. Self-Care

Mothers spoke of the daily activities they engage in for themselves. This theme was sub-divided into legitimacy, participation in health promoting activities, and spirituality.

#### 3.4.1. Legitimacy

Mothers expressed various levels of legitimacy regarding their needs, outside the mothering role. Some of the mothers stated that their own needs were either a low priority or not a priority at all:

‘Honestly, I don’t prioritize it’.(BC)

While some mothers viewed meeting their needs as a means to being a better caregiver:

‘I know it’s important. Because I know I have to fill up my reserves so I can do laundry, wash dishes, and make dinner. I know that if I do not give myself time, if I do not go out with friends, if I do not exercise a little—I will feel that I cannot function as a mother and as a good wife’.(YN)

However, two mothers expressed legitimacy of their own needs not related to their caregiver role:

‘I am a woman and person who is career-oriented, I have two degrees in biology, I work in research, in the field of fertility, I am trying to get accepted into medical school, I do lots of things, I mean I don’t give up on myself’.(Zh)

#### 3.4.2. Participation in Health Promoting Activities

Mothers expressed various degrees of active participation in health promoting activities, not related to their mothering roles, mainly in the domains of physical self-care, learning, and spirituality.

‘I make sure to sleep and I prepare myself food that I like and is tasty’.(Y)

‘I exercise, for example I walk to work instead of taking the bus so I can get in some exercise’.(R)

‘I watch lectures, it really “fills” me up’.(Zh)

‘I go to a lot of religious classes and a couple times a year I try to take a course in my profession’.(G)

However, almost all expressed that they wanted more participation in health promoting activities outside their mothering roles. They expressed their desire for it but that in the present stage of their lives they were unable because of the demands of the mothering role:

‘I take care of my basic need, like eating, sleeping and stuff, but I wish I had more time to do more’.(Z)

‘There is simply no time, there is no right time for it. There are all the other roles I previously told you—to take care of the home, to take care of the children, mainly these roles. And if there is some time then I need some rest and to sleep or eat. It is really depressing, that you want to do things and you do not find the time’.(BC)

#### 3.4.3. Spirituality

Religiosity was expressed as a backdrop to their participation in daily activities:

‘I live in spirituality; we live it in everything. Maybe physically you free up time for prayer but spirituality you live throughout the whole day, as someone who lives an ultra-orthodox lifestyle’.(T)

## 4. Discussion

The present study aimed to explore the lived experience of mothering a child with ADHD in the UOJ community. Findings of the present study shed light on the rich experiences of these mothers within the fabric of their cultural environment. The emerged themes reflect their challenges in raising a child with ADHD, their perceived centrality of the maternal role, the interaction of ADHD manifestations within their communities, and finally how they engage in health promoting self-care. As such the themes were: Child ADHD manifestations, maternal role, social factors, and self-care, which will be discussed in more detail below.

Participants in this study displayed a range of negative psychological health symptoms and distress, similar to that described of MoCwADHD in the extensive literature conducted across cultures and countries [6,8,9,10,14,30,33]. Mothers described the manifestations of their child’s ADHD, including difficulties staying focused and concentrating when talking to them, as well as in daily tasks, and their child’s challenges in regulating their emotions and behaviors according to social norms. Literature measuring behaviors associated with ADHD of participants in the UOJ community found that a large percentage of adolescents displayed symptoms that correspond to an ADHD diagnosis [19]. In addition, another study found that the percentage of UOJ children diagnosed with ADHD was similar to that in other countries [20]. Taken together, these findings reveal ADHD as a condition that also exists in this community, along with the challenges that may accompany it, and supports the literature that ADHD is a cross-cultural phenomenon [38]. However, an inclusion criteria for the study was a valid diagnosis of ADHD in the child, thus the rich descriptions of a child’s ADHD may have been better represented than among the general UO population. The prevalence of diagnosed ADHD in the UOJ community is lower than the general population, which may be related to mental health stigmas and shame that prevent parents going through the diagnostic process [17,18]. While ADHD is a cross-cultural condition, there are specific nuances regarding the community that emerged from the study’s findings which will be highlighted to further understand the cultural sensitives unique to this group.

Mothers described their perception of the maternal role in general and specifically in mothering a child with ADHD. All the mothers described motherhood as the most central role in their lives, and repeatedly stated themselves as being a “mother first”. Mothers used religious framing and contexts when describing their role. Motherhood in the UOJ community is considered an important role, a means for self-actualization, and a religious responsibility. It is connected to her experience of spirituality and relationship with God. Motherhood goes beyond raising children—it is a religious identity and conduit through which a woman fulfills her purpose in life. As such, children’s behavior is viewed as a reflection of the mother’s competency and success in her role [13]. The results of this study resonate with the literature on mothering as a contextual role. Expectations and beliefs regarding the mothering role are determined and learned from the mother’s socio-cultural environment. As such, the maternal role identity is developed through a process where the mother experiences competence in her mothering role, based on the belief of what is expected of her [12]. As mothering is such a significant role for a woman in this community and her child’s behavior is reflective of her success and identity in this role, there was a strong expression of its totality and her expectation to do whatever it takes to raise her child. This sense of totality was in the background when discussing the burden of raising a child with ADHD. The tension between the mother’s expectations of herself along with the specific manifestations of her child’s ADHD was expressed in feelings of incompetency. Mothers also expressed their child’s ADHD as a life-long struggle and the challenge of managing the needs of the child alongside the needs of other family members. Feelings of incompetency [10] and difficulty balancing the multiple needs of family members in the context of having a child with ADHD [8] is a shared experience among MoCwADHD in the general population. However, the specific cultural context of the mother in how it shapes her identity, fulfills her purpose in life, and influences her definition of competency and success in her role may provide insight into the uniqueness of her mothering experience.

Some mothers described the need to balance between their role as a mother and other roles, like a careerist. In the UOJ community, women are generally the main breadwinners and responsible for the economic management of the household so that their husbands can be available to study religious texts full-time. This may seem paradoxical, in that generally women in UO religions have an expectation to fulfill their domestic roles and may play less of a role in the work force. However, since the study of the religious texts by the men full-time is a high value to both men and women in this community, women have therefore become the breadwinners to support their families’ economic needs [13,39]. Participants in this study were all married, relatively highly educated, and working women in a variety of areas, mostly in the health professions. In response to the rise in the standard and cost of living, along with the economic distress among the UOJ community, women have begun to pursue higher education so they can better support their families’ economic needs. Many women have been seeking professions, like those in the health field, that enable them to maintain their UO identity while earning a wage that helps support their families [35,39]. However, since the mothering role holds a great value, there is a challenge experienced by mothers when trying to balance both motherhood and work requirements [13]. The challenge of balancing work and mothering is one shared by working mothers in the general population as well [40]. However, the culture-specific expectations and beliefs of the mothering role in the UOJ community, such as its influence on her religious identity and overarching purpose in life, may provide additional insight into the experience and challenge of balancing both roles for these mothers. Mothers in this study described how balancing work and mothering was very hard and that they had to pay a price when balancing these two roles, since both are demanding and integral to her role and responsibility as a woman in this community [41]. Additionally, it was interesting to note that while there is a cultural expectation of women to work, and a high value to mothering, many of the women expressed their jobs as a means of self-fulfillment and pleasure. Mothers first discussed their role as breadwinners in the context of their responsibilities for providing economic security for their families. Those who experienced success and competence in their working roles also expressed feelings of self-fulfillment and pleasure. This finding may be in line with universal psychological needs theories, whereby this working role seemed to meet some of their needs for belonging, competence, esteem, and self-actualization [42,43].

Mothers’ knowledge regarding ADHD was limited. Access to sources of information, including technology and social media, is more restricted in the UOJ community in response to their conservatism of outside influences. As such, parents often present with information not based on reliable professional sources, but rather based on informal sources from teachers who are not experts in ADHD, friends, and/or family members [18]. However, when presented with a diagnosis, mothers expressed the benefit of understanding their child’s behavior in terms of a medical problem, as opposed to a behavioral issue. Receiving a clear diagnosis for a child’s condition has been shown to help parents attribute their child’s behavior to a biological source, thereby reducing blame in themselves and their child [2,44], as was displayed in the mothers in this study as well.

Beyond knowledge of ADHD, mothers expressed using various resources and strategies when mothering their child with ADHD. These included pharmacological treatment, management skills, psychological framing, and avoidance, all of which can also be conceptualized as coping strategies. Coping can be defined as the cognitive and behavioral efforts one does to overcome, manage, tolerate, and/or reduce the external and internal demands and conflicts created by stressful situations that exceed the resources of the individual [45]. The techniques used to cope are known as coping strategies and can be categorized into three types. The first is problem-focused coping strategies, which allows one to manage a situation by modifying the problem. The use of pharmacological treatment and management skills are examples of such strategies [6,46]. The second is emotion-focused coping strategies, in which the individual tries to manage emotional tension because of the problem and which psychological re-framing can be an example. The third is avoidance-focused coping strategies, where one uses cognitive and behavioral efforts to deny, minimize, or avoid dealing with the stressful situation, and was described by some of the mothers as well [45,46]. Most of the mothers in this study stated problem-focused and emotion-focused coping strategies as their main strategies. Pharmacological treatment is used as the central problem-focused strategy for treating ADHD in the UOJ community [17]. It was strikingly noticeable that psycho-social interventions were missing from the narratives of the mothers’ experiences. The American Academy of Pediatrics’ (2019) clinical guidelines for ADHD recommends a combined approach in treating ADHD, including both pharmacological treatments along with parent-training and other psycho-social interventions [47], and which is commonly used in the general population [10]. It is possible that stigma around a medical diagnosis may have played a role in the limited use of psychosocial interventions among the UOJ mothers, since medication is easier to hide and therefore less visible [17]. Additionally, religious framing, an emotion-focused strategy which was used by participants, has also been used by mothers of children with other disabilities in the UOJ community to help make meaning of their challenge [44].

Mothers stated various social factors that impacted their experiences of mothering their child with ADHD. One of these factors included having support from their spouses, which emerged as an important determinant of mothers’ QoL. This finding is interesting considering the traditional context in the UOJ community, whereby women are mainly responsible for the home and finances, thus leading to the assumption that fathers are not involved. However, fathers often play a role in helping reduce some of their working wife’s domestic and childcare responsibilities. Despite the traditional model of women mothering and men studying, it is interesting to see this involvement of fathers. Typically, the fathers’ schedules are unique in that they have long afternoon breaks where they can support their wives with child-care and domestic errands [41]. This cultural context may further provide the conditions for promoting fathers to be more involved in raising children with ADHD.

Regarding the healthcare system, most of the mothers described not receiving professional attention and service. While this experience is shared with MoCwADHD in the general population [7], additional cultural related barriers may have affected their experience in receiving appropriate treatment for their child. Mothers in this study described difficulty communicating with medical professionals as well as recommendations that were not in line with the families’ cultural sensitivities. The UOJ community is one that is insular, cautious of outside influences and thus underrepresented in research. Healthcare workers may not be aware of the cultural sensitivities of this particular group. Community members prefer to be treated by professionals who share their religious values or professionals recommended by their rabbinical authority. For example, immodest dress and treatment by a health professional of the opposite gender may pose as barriers to receiving treatment [48]. Thus, the importance for understanding cultural sensitivities to enhance culturally competent practice is crucial in helping reduce health disparities and improving the quality of service provided by healthcare workers to members of this community [17].

The educational system was another influential social factor on mothers in this study. It was interesting to note that there were gender differences regarding mothers’ experiences in the educational system. Mothers of girls with ADHD stated that their daughters received support and professional services and expressed a general trust in the staff’s capabilities in meeting their child’s needs. However, mothers of boys expressed that the teachers had insufficient psycho-educational tools and strongly recommended their child be treated with medication. Mothers went insofar as to consider sending their sons to a school less in line with their religious educational ideologies to help meet their child’s educational needs. This may be explained in that girls’ schools receive funds from the Ministry of Education and generally comply with the requirements of its core required curriculum to help enable girls receive an education that will allow them eventually to financially support their families. Teachers are required to have a professional degree in education and participate in continuous learning. Thus, they may have more knowledge and psycho-educational tools surrounding ADHD. However, boy’s schools are private institutions not under the supervision of the Ministry of Education. Torah learning is considered foundational and central in male studies. The community prefers not to have outside influential factors that may impact their studies. Boys learn Torah in addition to a small portion of the secular curriculum to ensure they are literate and have a basic grasp of arithmetic and teachers generally are not trained and lack professional degrees [49,50]. As such, this may explain why teachers in boys’ schools present with inefficient knowledge and tools to help support children with ADHD. It is interesting to note that in the general population, girls are more often under-identified and untreated in comparison to boys [51]. The separation of genders in education among the UOJ community may be a supportive factor for girls to be identified and treated for their ADHD.

When mothers described taking care of oneself, it was usually expressed as a low priority, as is seen in the general population as well [52]. However, when they did speak of the legitimacy to do so, it was mostly as a means to help them perform their mothering roles. Women in the UOJ community have reported experiencing personal fulfillment when putting their families first. A mother’s role is defined according to her relational capabilities, especially in how she relates to her family [13]. Thus, viewing her ability to help meet the needs of her family as a high value may have come before the value of meeting her personal needs. When mothers did describe the activities that they engaged in outside of their mothering roles, the categories were mostly in taking care of their physical needs, learning, some leisure, and spiritual pursuits. Research on mothers of children with disabilities, including MoCwADHD in the general population, depicts limited participation in a variety of health-promoting activities. Taking care of physical needs and some participation in leisure are the more common activities that these mothers engage in [52]. Mothers in this study also described participating in these physical and leisure activities, however spirituality was a common example provided when describing ways in which they take care of themselves. Spirituality is an all-encompassing experience for members of the UOJ community as it guides their thoughts and behaviors in all areas of life [13]. In addition, it was interesting to note that many mothers in this study also expressed participation in activities that involved learning as a way in which they took care of themselves. The UOJ community values learning, and therefore members are encouraged to read and learn as a leisure activity from a young age [53]. While MoCwADHD present with a limited range of participation in health activities, findings in this study display how contextual factors can play a role on the specific activities encouraged and engaged in among different cultures. This may be an important factor when considering treatment and intervention for this population.

### Limitations

It is important to note the limitations present in this study. As this is a qualitative study, the number of participants is low and therefore may not be representative of the population for generalizations. In addition, the sample may be influenced by a sample bias stemming from the snowball and convenience sampling methods used. As the information in this study was based on the accounts from mothers who were interested in participating, it is possible that they may have had an agenda or may have been biased in the reporting of their experiences [7]. The presence of a valid child ADHD diagnosis was an inclusion criterion, and thus mothers may have presented with more awareness and knowledge surrounding their child’s needs than other mothers in the UOJ community. UOJ children have a lower prevalence of ADHD diagnosis than children in the larger population [54], and so the mothers may have represented a unique sub-group of the larger population of MoCwADHD in the UOJ community. In addition, as the interview was performed by a member of the UOJ community, this may have influenced the interviewees’ responses. While it is known that members of the UOJ community are more cautious of health professionals outside the community [20], there is still some fear of stigma that may have been present on the part of the interviewee. The first author tried to provide a non-judgmental environment during the interview process, however this may have influenced the interviewees’ responses. Additionally, the study was conducted in Israel and therefore may not be representative of UOJ communities in other parts of the world. We therefore recommend future research exploring the lived experiences of mothers in the UOJ community by interviewers not from the community, including orthodox representatives of other religions, and in other countries as well.

## 5. Conclusions

The burden of mothering a child with ADHD in the UOJ community has been shown to negatively impact a mother’s psychological health and QoL. These health outcomes were shown to partially converge with those of the general population, such as ADHD manifestations. However, mothers’ experiences also diverge from the general population, demonstrating the unique impact of the UOJ socio-cultural environment in which she lives. These findings may help support cultural competency among healthcare workers by providing knowledge and bringing awareness to the need for culturally sensitive health care and interventions for this growing and under-served community.

## Figures and Tables

**Figure 1 ijerph-19-14483-f001:**
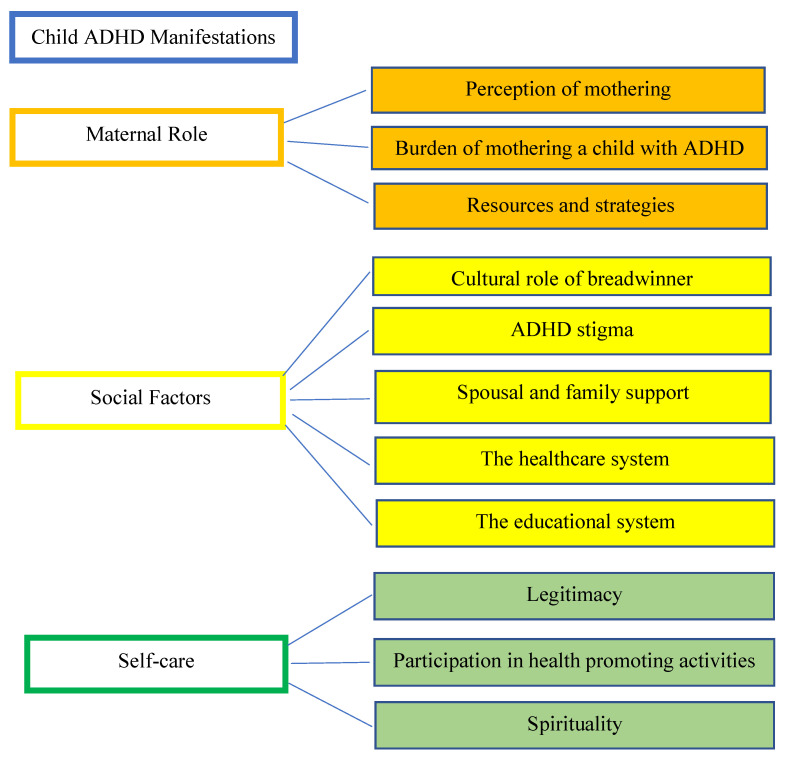
Themes and sub-themes obtained from interviews.

**Table 1 ijerph-19-14483-t001:** Socio-demographics of participants (*n* = 10).

Mother	Mother Age (yrs)	# Children	# Children with ADHD	Child Gender	Child Age	Mother’s Education	Employment	Family Income
G	43	6	2	f	16	Degree	Speech therapist	Above Avg
T	37	6	3	m	15	Degree	Vision therapist	Avg
M	39	5	1	f	16	High School	Saleswoman	Above Avg
R	37	6	1	m	9	Degree	Teacher	Below Avg
YN	38	4	1	m	6	Degree	Teacher	Below Avg
H	40	6	1	m	12	Primary	Saleswoman	Below Avg
Zh	31	3	1	m	6	Degree	Nurse	Avg
BC	30	3	1	m	6	Degree	Occupational therapist	Avg
Z	39	4	2	m	15	Degree	Administrative director	Below Avg
Y	45	6	1	m	18	High School	Secretary	Avg

Notes: (yrs) refers to age in years; # refers to number; Family income based on the national average. Avg = Average.

**Table 2 ijerph-19-14483-t002:** Participants’ characteristics regarding self-reported parenting stress, psychological symptoms, quality of life, and severity of child ADHD symptoms (*n* = 10).

Mother	PSI	PHQ-4	WHOQoL-BREF Pys	WHOQoL-BREF Psych	WHOQoL-BREF Social	WHOQoL-BREF Environ	C-ADHD Index
G	30	6	59	48	59	60	63
T	30	0	62	80	72	62	69
M	35	2	57	51	59	66	86
R	28	4	73	72	75	72	45
YN	34	7	57	56	64	58	67
H	31	4	57	64	69	70	64
Zh	32	2	78	69	80	68	67
BC	31	12	39	35	48	54	73
Z	30	5	62	53	37	50	54
Y	30	1	75	80	69	76	88

Notes: PSI, possible range—11–40, higher score indicates a greater level of stress; PHQ-4, possible range—0–12, higher score indicates lower psychological health; WHOQoL-BREF, higher score indicates greater level of QoL; C-ADHD Index, Conners, T-scores, higher score indicates more severe symptomatology.

**Table 3 ijerph-19-14483-t003:** Semi-structured qualitative interview.

Topic	Question
Mother’s perception of having a child with ADHD	⚬ Please describe the manifestations of your child’s attention deficit disorder. ⚬ Do you have more than one child in a home diagnosed with ADHD? ⚬ What are your child’s strengths and weaknesses? ⚬ Can you describe what it is like to be a mother of a child with ADHD? ⚬ What is it like to mother a child with ADHD in the ultra-orthodox community and specifically the community that you identify with?
Life roles	⚬ Tell me about your day-to-day life, what does a typical day look like? ⚬ How does your child’s ADHD affect your daily life activities? ⚬ What roles do you participate in your life? (e.g., mother, partner, worker, self-care) ⚬ Tell me about your experiences in these roles (For each position separately) ⚬ What is important to you? ⚬ What do you enjoy? ⚬ What are you good at? ⚬ What is challenging for you? ⚬ To what extent have you made changes in your participation due to your child’s ADHD?
Environments	⚬ Tell me about where you live/work. ⚬ Tell me about your community. ⚬ How is the atmosphere at home? ⚬ Tell me about other environments in your life (social, leisure, extended family) ⚬ How do your environments relate to ADHD and being a mother to a child with ADHD? ⚬ Is the environment supportive, enabling, neutral, restrictive?
What mother feels she needs	⚬ What do you need to promote your health and quality of life?
Concluding question	‘You have shared a lot, thank you for that’ Is there anything else you would like to add in addition to what we have spoken about?

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
