# Peer review of "Mothering a Child with ADHD in the Ultra-Orthodox Community"

_ijerph, 2022, doi:10.3390/ijerph192114483_

Round 1

Reviewer 1 Report

The article is interesting. A few suggestions for the text come to mind:

 1.I would dispense with the quotation in the title, after all we are dealing with a scientific article. 

2.Why are words like 'Maternal roles' capitalized in the abstract.

3.In the keywords, I think ADHD should also be included.  

4.The author writes in line 57 "UO Jews". Should this fact not be included in the abstract.

5.It is difficult to find in the text exactly when the research was carried out.

6.In line 353 the author briefly explains the role of women as "the main breadwinner". Shouldn't there be at least a little more in line 552. 

7.In the limitations, it would be worth noting that the results are not representative due to the number of people surveyed.

8. When indicating further research, it would be worth pointing out that it should also include orthodox representatives of other religions.

9. There is a suggestion to state, very generally, where the research was conducted. I think this too is a limitation.

Reviewer 2 Report

see attached file

Round 2

Reviewer 2 Report

Changes have amended the problems previously detected